# Training Large Language Models for Retrieval-Augmented Question Answering through Backtracking Correction

**Huawen Feng, Zekun Yao, Junhao Zheng, Qianli Ma**[*]
School of Computer Science and Engineering, South China University of Technology, China
541119578@qq.com, qianlima@scut.edu.cn

## Abstract

Despite recent progress in Retrieval-Augmented Generation (RAG) achieved by large language models (LLMs), retrievers often recall uncorrelated documents, regarded as "noise" during subsequent text generation. To address this, some methods train LLMs to distinguish between relevant and irrelevant documents using labeled data, enabling them to select the most likely relevant ones as context. However, they are susceptible to disturbances, as LLMs can easily make mistakes when the chosen document contains irrelevant information. Some approaches increase the number of referenced documents and train LLMs to perform stepwise reasoning when presented with multiple documents. Unfortunately, these methods rely on extensive and diverse annotations to ensure generalization, which is both challenging and costly. In this paper, we propose **Backtracking Correction** to address these limitations. Specifically, we reformulate stepwise RAG into a multi-step decision-making process. Starting from the final step, we optimize the model through error sampling and self-correction, and then backtrack to the previous state iteratively. In this way, the model's learning scheme follows an easy-to-hard progression: as the target state moves forward, the context space decreases while the decision space increases. Experimental results demonstrate that **Backtracking Correction** enhances LLMs' ability to make complex multi-step assessments, improving the robustness of RAG in dealing with noisy documents. Our code and data are available at https://github.com/201736621051/BacktrackingCorrection.

## 1 Introduction

Large language models (LLMs)(Zhao et al., 2023) are constrained by the knowledge they acquired during training, meaning they may lack up-to-date or specialized information on certain topics(Zhang et al., 2023; Li et al., 2023). Retrieval-Augmented Generation (RAG)(Guu et al., 2020), which retrieves relevant information from external sources (e.g., Wikipedia) before generating responses, is an effective approach to address this limitation, particularly for knowledge-intensive tasks(Lewis et al., 2020). However, these methods are highly context-dependent, and their performance can significantly deteriorate if the retrieved documents are irrelevant to the topic at hand (Shi et al., 2023b). Additionally, they are prone to being misled by noisy documents, potentially leading to inaccurate outputs (Mallen et al., 2023a).

Early methods improve the robustness of LLMs through counterfactual data augmentation (Neeman et al., 2023; Yoran et al., 2023). For each triplet of question, relevant document, and answer $(Q, D^R, A)$, the document $D^R$ is perturbed and transformed into an irrelevant document $D^I$. In addition to training on the original data $(Q, D^R, A)$, the model is also trained on the counterfactual data $(Q, D^I, A)$, forcing it to maintain the original output $A$ even when presented with $D^I$. However, while these methods help disentangle the contextual and parametric knowledge of LLMs, they remain limited in determining when to apply each type of knowledge when dealing with unlabelled documents retrieved by the retriever (Shi et al., 2023a), particularly when the two knowledge sources

---

[*]Corresponding author.

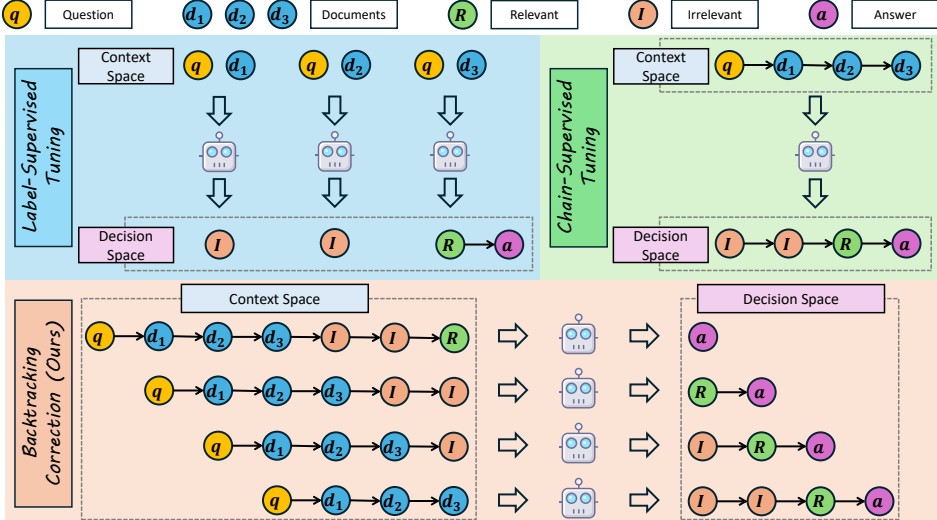

Figure 1: The comparison between Label-Supervised Tuning (LST), Chain-Supervised Tuning (CST) and Backtracking Correction. Label-Supervised Tuning trains the LLMs to assess retrieved documents separately. Chain-Supervised Tuning trains the model with entire reasoning paths. Backtracking Correction introduces the backtracking algorithm to simplify the learning for reasoning chains.

lead to different answers. The model is still unable to assess whether the retrieved documents are relevant or whether they should be referenced.

Considering that, several methods have been developed to enhance the self-reflection capabilities of LLMs. SELF-RAG (Asai et al., 2023), for example, trains a self-reflective LLM that retrieves passages as needed and reflects on their content. Similarly, REAR (Wang et al., 2024c) trains LLMs to evaluate the relevance of documents to improve self-awareness, while FILCO (Wang et al., 2023) trains context-filtering models that can sift through retrieved contexts. These approaches rely on label-supervised training data, where each document is assigned a simple binary label (Label-Supervised Tuning), keeping annotation costs low. However, the sparse supervision in the decision-making process limits the model's ability to understand why a retrieved document is relevant or irrelevant (Wang et al., 2024a). Moreover, they cannot fully eliminate noisy documents, as LLMs remain prone to errors when the top-ranked document is noisy.

Recent studies have increased the number of referenced documents and employed LLMs to perform stepwise analysis. Inspired by Chain-of-Thought (CoT)(Wei et al., 2022), Yu et al. (2023) introduced Chain-of-Note (CON), which generates sequential reading notes for retrieved documents step by step. Similarly, RAFT(Zhang et al., 2024) utilizes chain-of-thought-style responses as supervision, training LLMs to disregard distractor documents. Unlike Label-Supervised Tuning, these methods use reasoning chains as supervision (Chain-Supervised Tuning), offering more detailed feedback during training. However, stepwise-annotated data is challenging to collect, and much of it depends on proprietary language models (e.g., ChatGPT, GPT-4). Moreover, the complexity of reasoning chains (decision space) makes them difficult for LLMs to learn, requiring extensive and diverse annotations to ensure generalization (Gülçehre et al., 2023; Xie et al., 2023).

The challenges mentioned above have motivated us to propose **Backtracking Correction**. Specifically, we reformulate RAG into a multi-step decision-making process. As illustrated in Figure 1, given a query and three referenced documents, LLMs evaluate each document sequentially and ultimately provide an answer. Drawing inspiration from Step-DPO (Lai et al., 2024) and ReFT (Trung et al., 2024), we first warm up the model with SFT, then apply stepwise preference optimization to reasoning chains. Starting from the final state, we sample model-generated errors and employ self-correction to optimize the current state through Reinforcement Learning (RL), then backtrack to the previous state. Unlike Chain-Supervised Tuning which learns the entire reasoning path, our method optimizes each step by focusing on decision-making based on the current state, as the remaining steps have already been learned. This approach gradually reduces the context space while expanding the decision space,

following an easy-to-hard progression. In essence, **Backtracking Correction** simplifies the learning of reasoning chains in RAG and eliminates the need for external annotations. The main contributions of this paper are summarized as follows:

- We identify the limitations of Label-Supervised Tuning (LST) and Chain-Supervised Tuning (CST) through detailed comparisons.

- We transform RAG into a multi-step decision-making process and introduce **Backtracking Correction** to more effectively train the reasoning abilities of Retrieval-Augmented Language Models (RALMs).

- Extensive experiments demonstrate the effectiveness of **Backtracking Correction**, with ablation and comparison studies explaining how and why it works.

## 2 RELATED WORK

### 2.1 NOISE ROBUSTNESS IN RAG

Noise robustness refers to a retrieval-augmented language model's ability to discern and disregard noisy information present in retrieved documents while effectively leveraging its intrinsic knowledge (Chen et al., 2024a). State-of-the-art RAG architectures (Asai et al., 2023; Wang et al., 2024c) suggest that LLMs can learn to utilize external knowledge adaptively to defend against attacks of noisy information. They train the LLM to evaluate retrieved documents and use the highest-ranking ones to answer questions. Unfortunately, these methods process retrieved documents separately and lack comparisons, making it easy to overlook relevant information. To address this, some approaches (Zhang et al., 2024; Yu et al., 2023) employ LLMs to assess all referenced documents within the same context. However, the complex reasoning chains (decision space) can be challenging for LLMs to learn.

### 2.2 FINE-TUNING BASED ON COT

Current approaches to solving reasoning tasks utilize Supervised Fine-Tuning (SFT) to train LLMs using Chain-of-Thought (CoT) annotations (Wang et al., 2024b). However, each sample in the training data typically contains only one annotated reasoning path, which hinders generalization due to the potential for multiple interpretations of the same documents. To address this, some studies have adopted Reinforcement Learning to enhance performance beyond SFT. Trung et al. (2024) samples various CoT reasoning paths and applies Proximal Policy Optimization (PPO) to learn from them. Lai et al. (2024) treats individual reasoning steps as separate units to provide fine-grained supervision for preference optimization. Nevertheless, the training data requires extensive annotations, significantly increasing costs.

### 2.3 SELF-PLAY IN REINFORCEMENT LEARNING

Self-play describes a type of multi-agent learning that involves deploying an algorithm against copies of itself to test compatibility in various stochastic environments (DiGiovanni & Zell, 2021). The broader goal is to transform weak models into strong ones without the need for additional human-annotated data. Chen et al. (2024b) propose Self-Play Fine-Tuning (SPIN), where the LLM refines its capabilities by competing against its instances. Unfortunately, while this method maximizes the potential of a limited amount of training data, it still requires high-quality annotations as the final target.

## 3 METHODOLOGY

### 3.1 PRELIMINARIES

Consider a high-quality question answering dataset $S_{LS} = \{(q, D, L, a)\}_{i=1}^{n}$ where $q$ is the question, $a$ is the answer, and $D = [d_1, d_2, ..., d_T]$ are several relevant and irrelevant documents labelled with $L = [l_1, l_2, ..., l_T]$, we can employ Label-Supervised Tuning (LST) to train the LLMs to distinguish

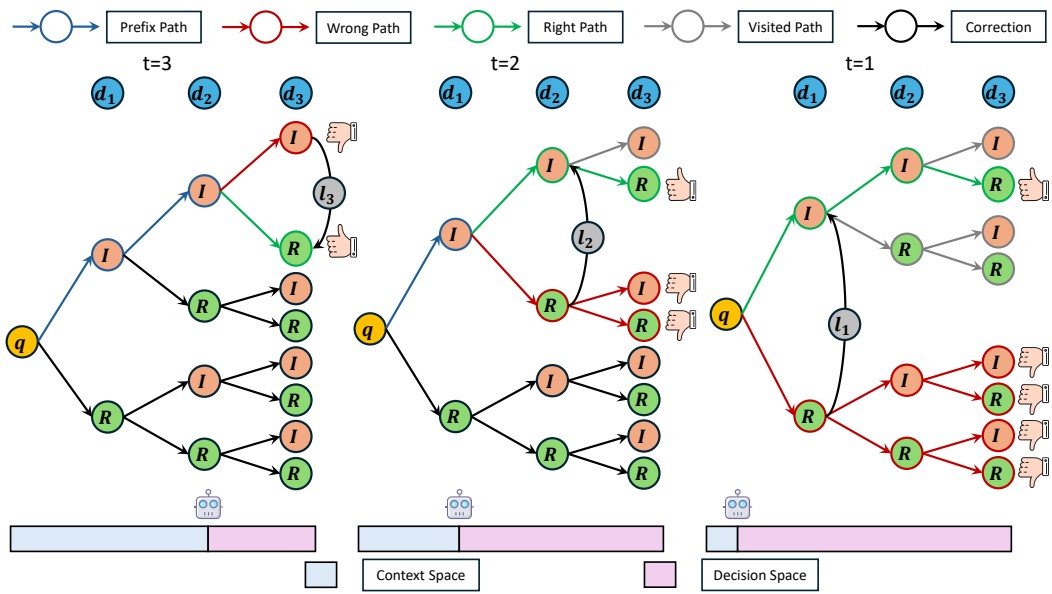

Figure 2: Backtracking Correction begins optimization from the last state and iteratively backtracks to the previous states. Each optimization step focuses solely on decision-making at the current state, as the preceding states have already been optimized - meaning the remaining decision space has been explored. This approach enables the model to correct self-generated errors and differentiate between correct and incorrect chains at each state, thereby enhancing its reasoning ability to assess the relevance of given documents.

the documents separately:

$$\mathcal{L}_{LS} = -\sum_{i=1}^{T} l_i \log\left(\Pi_\theta(q, d_i)\right) + (1 - l_i)\log\left(1 - \Pi_\theta(q, d_i)\right) \tag{1}$$

where $\theta$ indicates the LLM's parameters and $\Pi_\theta()$ represents the generating process of the model.

Integrating the reasoning chains $C$ annotated by human or AI, the dataset can be represented as $S_{CS} = \{(q, D, L, C, a)\}_{i=1}^{n}$ and we can conduct Chain-Supervised Tuning based on it:

$$\mathcal{L}_{CS} = -\sum \log P(\Pi_\theta(C, a|q, D)) \tag{2}$$

However, collecting annotations for $C$ is costly. Additionally, learning the entire reasoning chains necessitates a large amount of training data to achieve effective generalization. To address this, we introduce a new fine-tuning method - Backtracking Correction - to enhance the reasoning performance of RALMs without requiring additional annotations or feedback.

### 3.2 TASK FORMULATION

Given a question $q$ and a list of retrieved documents $D$, the LLM needs to assess them one by one and provide the reasoning chains $C = [c_1, c_2, ..., c_T]$ before answering the question, including the final judgement and its corresponding explanation. Regarding the task as a multi-step decision-making process, the reasoning process can be represented as:

$$s_i = \begin{cases} q, D & if \quad i = 0 \\ q, D, c_1, c_2, ..., c_i & if \quad i \in [1, T] \end{cases} \tag{3}$$

$$c_t \sim \Pi_\theta(s_{t-1}) \quad a \sim \Pi_\theta(s_T) \quad t \in [1, T] \tag{4}$$

where $c_t$ is the decision made by the LLM at state $t$ (the reasoning result for document $d_t$) and $a$ is the final answer generated by the LLM.

## 3.3 BACKTRACKING CORRECTION

Inspired by ReFT (Trung et al., 2024), we conduct CoT sampling to collect the errors in the reasoning chains. By dividing the reasoning result $c_t$ at each state $t$ into "relevant" and "irrelevant", we can get a binary tree. As shown in Figure 2), each node represents a specific decision regarding the document, while each path denotes a reasoning chain. It is important to note that although each node (decision) is unique, it may correspond to multiple explanations, as the LLM can generate different content that leads to the same decision. Here, we only consider whether the final judgment aligns with the document's label (relevant or irrelevant). In this way, we may obtain the wrong chains generated by the LLM.

As shown in Equation 4, the previous steps $c_1, c_2, ..., c_{t-1}$ can be viewed as examples of in-context learning for $c_t$. In other words, the output at state $t$ can be influenced by prior reasoning results, making it difficult to identify which step caused the error at the current state. To address this, we control the length of the correct prefix (context space) to limit the steps where errors may occur (decision space). By feeding a completely correct prefix $s_t^+$ into the model, we can ensure that none of the errors in the model's output $o_t$ are caused by previous states:

$$s_t^+ = [q, D, c_1, c_2, ...., c_t] \quad o_t = [c_t, c_{t+1}, ...., c_T, a]$$
$$\forall c_i \in s_t^+ True(c_i) \quad \exists c_i \in o_t False(c_i) \quad o_t \sim \Pi_\theta(s_{t-1}^+) \tag{5}$$

where $True(c_i)$ indicates $c_i$ is correct while $False(c_i)$ means it is wrong.

In this way, we can collect the on-policy errors $o_t$ with corresponding correct input $s_{t-1}^+$ at different state $t$. We then employ Self-Correction to optimize the LLM at each state $t$ using Reinforcement Learning (RL). Based on the Self-play mechanism (DiGiovanni & Zell, 2021), the method conducts multi-turn training on self-corrected data to avoid distribution mismatches between the original and corrected chains.

Consider a two-player game: Player A's objective is to distinguish between incorrect and corrected chains, while Player B aims to correct the wrong chains and generate the right ones in a manner indistinguishable from the on-policy errors. Specifically, Player B is the old LLM from the previous turn, while Player A is the LLM being trained in the current turn. The learning process involves generating corrected outputs with Player B, optimizing Player A with both original and corrected outputs, employing Player A as a reward to optimize Player B, and generating new outputs with Player B:

$$... \Rightarrow B_\theta(s_{t-1}^+, o_t^-) \rightarrow o_t \Rightarrow \nabla_\theta L_A(s_{t-1}^+, o_t^-, o_t) \Rightarrow \nabla_\theta L_B(s_{t-1}^+, A) \Rightarrow B_\theta(s_{t-1}^+, o_t) \rightarrow o_t^+ \Rightarrow ... \tag{6}$$

where $L_A$ and $L_B$ are objectives of Player A and Player B.

**Player A**'s objective is to distinguish between incorrect and corrected chains without relying on gold labels. Therefore, we aim to maximize the gap between the reward values of the distributions of the two chains:

$$\arg\max_\theta \mathbb{E}_{s_{t-1}^+, o_t \sim \Pi_\theta(s_{t-1}^+), o_t^+ \sim \Pi_\theta(s_{t-1}^+, o_t, L)} \left[ r(s_{t-1}^+, o_t^+) - r(s_{t-1}^+, o_t) \right] \tag{7}$$

where $r(\mathbf{x}, \mathbf{y})$ is the reward function of input $\mathbf{x}$ and output $\mathbf{y}$. $o_t^+$ is the corrected chains generated by the LLM fed with original reasoning chains and the gold labels $L$.

To avoid unbounded reward gap, we utilize the logistic function $\sigma$ to replace Equation 7:

$$\arg\min_\theta -\mathbb{E}_{s_{t-1}^+, o_t \sim \Pi_\theta(s_{t-1}^+), o_t^+ \sim \Pi_\theta(s_{t-1}^+, o_t, L)} \left[ \log \sigma(r(s_{t-1}^+, o_t^+) - r(s_{t-1}^+, o_t)) \right] \tag{8}$$

**Player B** aims to correct $o_t$ with $L$ and generate new reasoning chains within the original distributions to achieve a high score from Player A. Player B must avoid distribution mismatch, as out-of-distribution correct chains may not effectively enable the model to correct its own mistakes (Kumar et al., 2024). By incorporating a Kullback-Leibler (KL) regularization term, we can formulate the objective function for Player B:

$$\arg\max_\theta \mathbb{E}_{s_{t-1}^+, o_t \sim \Pi_\theta(s_{t-1}^+), o_t^+ \sim \Pi_\theta(s_{t-1}^+, o_t, L)} \left[ r\left(s_{t-1}^+, o_t^+\right) \right]$$
$$- \beta \mathbb{D}_{KL} \left[ \Pi_\theta(o_t^+ | s_{t-1}^+) || \Pi_{ref}(o_t^+ | s_{t-1}^+) \right] \tag{9}$$

where $\beta$ is the parameter controlling the deviation from the base model.

Following the previous work (Peters & Schaal, 2007; Peng et al., 2019), we can get the optimal solution to Equation 9:

$$\Pi_\theta(o_t^+|s_{t-1}^+) = \frac{1}{S(s_{t-1}^+)}\Pi_{ref}(o_t^+|s_{t-1}^+)\exp(\frac{1}{\beta}r(s_{t-1}^+, o_t^+)) \tag{10}$$

where $S(s_{t-1}^+)$ is a function of only $s_{t-1}^+$ and $\Pi_{ref}$.

It is worth noting that $o_t^+$ is generated based on $o_t$ while prompting Player B with gold labels. Similarly, $o_t$ is constructed based on the former output $o_t^-$. Hence, we can get the optimal solution for Player B in the previous turn:

$$\Pi_\theta(o_t|s_{t-1}^+) = \frac{1}{S(s_{t-1}^+)}\Pi_{ref}(o_t|s_{t-1}^+)\exp(\frac{1}{\beta}r(s_{t-1}^+, o_t)) \tag{11}$$

Based on Equation 10 and Equation 11, we can calculate $r(s_{t-1}^+, o_t^+)$ and $r(s_{t-1}^+, o_t)$:

$$r(s_{t-1}^+, o_t^+) = \beta\log\frac{\Pi_\theta(o_t^+|s_{t-1}^+)}{\Pi_{ref}(o_t^+|s_{t-1}^+)} \quad r(s_{t-1}^+, o_t) = \beta\log\frac{\Pi_\theta(o_t|s_{t-1}^+)}{\Pi_{ref}(o_t|s_{t-1}^+)} \tag{12}$$

Substituting Equation 12 into Equation 8, we can get the end-to-end loss function:

$$\mathcal{L}_{BC} = \arg\min_\theta -\mathbb{E}\left[\log\sigma(\beta\log\frac{\Pi_\theta(o_t^+|s_{t-1}^+)}{\Pi_{ref}(o_t^+|s_{t-1}^+)} - \beta\log\frac{\Pi_\theta(o_t|s_{t-1}^+)}{\Pi_{ref}(o_t|s_{t-1}^+)}\right] \tag{13}$$

The final loss function $\mathcal{L}_{BC}$ looks similar to the policy objective of Direct Preference Optimization (DPO). However, DPO relies on pairwise annotations while Backtracking Correction does not. Backtracking Correction requires only a small SFT dataset $S_{CS}$ to equip the LLM with self-correction capabilities, followed by training on $S_{LS}$ without additional annotations. Consequently, the training data and reference models in Backtracking Correction are dynamic, while those in DPO are static, highlighting another notable difference.

Based on Equation 13, the LLM is trained on the data that includes $s_{t-1}^+$, $o_t^+$, and $o_t$. In other words, Backtracking Correction optimizes the decision space of the LLM at state $t$. Notably, $t$ is negatively correlated with the complexity of the decision space: the larger $t$ is, the smaller the decision space becomes, making optimization easier. Given that, we begin training from the end state $s_{T-1}$ to avoid encountering a complicated decision space and challenging optimization from the start of training. After optimizing the state $s_{T-1}$, we backtrack to $s_{T-2}$, sampling wrong output at $s_{T-2}$ for training. This process is repeated iteratively until all paths in the decision space have been explored. The optimization at each step focuses solely on the current state, as the remaining states have already been optimized. Algorithm 1 outlines the complete training procedure.

---

**Algorithm 1: Training Process**

**Input:** Large language model $\Pi_\theta$, chain-supervised dataset $S_{CS}$, label supervised dataset $S_{LS}$.
Initialize $\Pi_\theta$ with $L_{CS}$ (Equation 2) on $S_{CS}$.
**for** $t = T-1, T-2, ..., 1$ **do**
    Sample the errors in reasoning chains at current state $t$: $o_t \sim \Pi_\theta(s_{t-1}^+)$.
    Employ $\Pi_\theta$ to correct $o_t$ with gold labels $L$ from $S_{LS}$: $o_t^+ \sim \Pi_\theta(s_{t-1}^+, o_t, L)$.
    Train $\Pi_\theta$ with $\mathcal{L}_{BC}$ (Equation 13) on $s_{t-1}^+$, $o_t^+$, and $o_t$.
**end**
**return** $\Pi_\theta$

---

## 4 EXPERIMENTS

### 4.1 EXPERIMENTAL SETTINGS

#### 4.1.1 DATASETS AND BACKBONES

To ensure a fair comparison, we adopt the settings as consistent as possible with the previous work (Asai et al., 2023; Wang et al., 2024c). We use KILT (Petroni et al., 2021) to collect training

data $S_{LS}$ and utilize Wikipedia as the knowledge base. As mentioned earlier, we require a small SFT dataset $S_{CS}$ to initialize our model. For this, we use Llama3-8B-Instruct (Dubey et al., 2024) to obtain the original annotations for $S_{CS}$. The prompt to correct the wrong answer is:

> Summarize the document briefly and explain why it is **{relevant/irrelevant}** to the question. Please end your answer with 'So, the document is **{relevant/irrelevant}** to the question'.
> **{Query}**
> **{Document}**

It is important to note that the LLM learns meta-skills of self-correction on $S_{CS}$, rather than final reasoning ability, similar to the initial learning phase of Self-Evolution (Tao et al., 2024). The difference is that Backtracking Correction does not need to develop self-feedback abilities because $S_{CS}$ contains $L$, which can serve as feedback. In fact, the LLM is learning to summarize the consistent or conflicting points in the documents based on the query, a relatively straightforward task that does not require strong annotators. For the backbones, we use Llama2 7B (Touvron et al., 2023) as the LLM and Contriever (Izacard et al., 2022) as the retriever, both of which align with previous work (Asai et al., 2023; Wang et al., 2024c). Additionally, we also apply our method to Llama3 8B. For evaluation, we utilize four open-domain question-answering datasets: Pop QA (Mallen et al., 2023b), Trivia QA (Joshi et al., 2017), Web Questions (Berant et al., 2013), and SQuAD (Rajpurkar et al., 2016). The prompt for training is listed below:

> Given the question: **{Query}**
> You are provided with several documents.
> **{Documents}**
> Your task is to analyze each document one by one and determine if the document is relevant to the question. Please follow the steps below for each document:
> Your task is to analyze each document one by one and determine if the document is relevant to the question.
> After analyzing all documents, provide a final answer to the question based on the analysis of the documents starting with 'Final Answer: '.

### 4.1.2 BASELINES

**Base Models.** They are the foundational, pre-trained models that serve as the core for further fine-tuning or adaptation to code tasks. The base models include LLaMA2-7B-base Touvron et al. (2023) and LLaMA3-8B-Instruct Dubey et al. (2024).

**Proprietary Models.** These LLMs, unlike open-source models, are developed, owned, and managed by a private entity or organization. They are trained on specialized or private datasets that are not publicly available to serve specific business needs or objectives. Access to these models is usually based on API calls. The proprietary models include ChatGPT (Ouyang et al., 2022) and GPT-4 (OpenAI et al., 2024).

**Fine-tuned Models.** We also make comparisons with strong open-source LLMs. These models are pre-trained and fine-tuned, and aligned with humans, enabling them to follow the instructions accurately. For the sake of fairness, we adopt the system prompt used during the training stage. These LLMs will be given or not given the retrieved documents to implement the baselines with or without the retrieval. The fine-tuned models include ChatGLM3-6B (Zeng et al., 2022; Du et al., 2022), Toolformer-6B (Schick et al., 2023), Mistral-7B (Jiang et al., 2023), BaiChuan2-7B-chat (Yang et al., 2023), LLaMA2-7B-chat, Alpaca-7B (Taori et al., 2023), and some strong RAG models like RobustLM-7B, Self-RAG-7B, and REAR.

### 4.1.3 TRAINING AND INFERENCE SETTINGS

The training is conducted on Alignment Handbook. We use DeepSpeed zero stage 3 (Rajbhandari et al., 2020) to conduct distributed training on 8 A800 GPUs with 80GB memory. FlashAttention (Dao et al., 2022) is also applied to improve the training efficiency. We set the learning rate to 2e-5 and 1e-6 and the epochs to 2 and 1 for stage 1 and stage 2, respectively. The warm-up ratio is configured at 0.1, the global batch size is 128 and $T$ is set to 5.

|  |  | Pop QA | Trivia QA | Web Questions | SQuAD |
|---|---|---|---|---|---|
| Proprietary | ChatGPT | 24.7 | 78.2 | 57.0 | 28.5 |
|  | GPT-4 | 38.7 | 83.4 | 61.6 | 35.1 |
| Base | LLaMA2-7B-base | 4.5 | 10.9 | 6.1 | 3.7 |
|  | LLaMA3-8B-base | 5.7 | 10.8 | 4.4 | 3.8 |
| Fine-tuned w/o Retrieval | ChatGLM3-6B | 9.5 | 19.6 | 14.9 | 4.3 |
|  | Mistral-7B | 25.2 | 66.1 | 56.8 | 25.9 |
|  | BaiChuan2-7B-chat | 25.7 | 40.7 | 38.6 | 13.1 |
|  | LLaMA2-7B-chat | 25.1 | 58.7 | 48.6 | 19.1 |
|  | Alpaca-7B | 25.6 | 49.4 | 39.6 | 14.1 |
| Fine-tuned w/ Retrieval | ChatGLM3-6B | 42.6 | 35.7 | 24.3 | 19.5 |
|  | ToolFormer-6B | - | 48.8 | 26.3 | 33.8 |
|  | Mistral-7B | 59.1 | 71.2 | 57.6 | 42.0 |
|  | BaiChuan2-7B-chat | 56.2 | 63.3 | 50.0 | 38.7 |
|  | LLaMA2-7B-chat | 54.0 | 63.1 | 45.3 | 37.2 |
|  | Alpaca-7B | 54.2 | 58.6 | 47.3 | 32.6 |
|  | Self-RAG-7B | 53.8 | 62.6 | 32.4 | 26.5 |
|  | REAR | 52.9 | 71.6 | 38.4 | 43.8 |
|  | *RobustLM-13B* | *49.1* | *62.0* | *27.3* | *27.4* |
| Backtracking Correction | +LLaMA2-7B-base | 58.4 | 68.1 | 47.6 | 44.5 |
|  | +LLaMA3-8B-base | 59.3 | 70.1 | 49.3 | 44.9 |

Table 1: Exact match of all the baselines on four open-domain question answering datasets.

During inference, we use vllm (Kwon et al., 2023) to speed up inference. For all baselines, the temperature is set to 0.8, and the cumulative probability of the top tokens is set to 0.95. We evaluate LLMs' final performance based on whether their generation contains gold answers (Asai et al., 2023).

## 4.2 MAIN RESULTS

Table 1 shows the overall results on four open-domain QA datasets. To make a fair comparison, for all the ranking methods, the number of retrieved documents is set to 5 for all ranking methods. **In general, fine-tuned LLMs with retrieval perform better than those without, particularly on tasks involving their knowledge blind spots.** Nearly all LLMs struggle to directly answer questions in Pop QA, with accuracy rarely exceeding 26%, including the strong proprietary LLM, ChatGPT. However, their performance are significantly improved when provided with retrieved documents. Almost all achieving an accuracy above 50%, underscoring the importance of RAG. **Conversely, RAG may not be as beneficial when LLMs possess the relevant knowledge.** For example, the accuracy of LLaMA2-13B-chat without external documents on Web Questions is 52%, but it drops to 47.1% when augmented by retrieval. This phenomenon suggests that retrieval can negatively impact LLMs in areas where they are already proficient, as it may introduce additional noise.

Compared to strong fine-tuned LLMs and expert RAG models, Backtracking Correction outperforms most of them across the four datasets. Notably, chat models are instruction-tuned on massive external data, and RAG models require annotations from proprietary LLMs. In contrast, Backtracking Correction does not rely on extra data or annotations. Instead, it trains the model in a self-generated and self-improved manner, which demonstrates the advantages of our approach. **These results indicate that it is possible to achieve competitive performance by further leveraging the original dataset, rather than relying on additional data and annotations.**

## 4.3 BACKTRACKING CORRECTION V.S. OTHER TRAINING METHODS

To enable a fairer comparison with other training methods, we employ Label-Supervised Tuning (LST), Multi-document Label-Supervised Tuning (MLST), Chain-Supervised Tuning (CST), and Backtracking Correction (BC) to train the same LLM (Llama2 7B) on the same training data ($S_{LS}$ and $S_{CS}$ used by BC), eliminating the impact of additional data and annotations on performance. In particular, MLST trains the LLM on $S_{LS}$ similarly to LST, but its input consists of multiple documents rather than a single document. Table 2 presents a case where LST, MLST, CST, and BC produce different answers based on the same documents. LST assigns the highest score to the

Who is the author of Fruits?
Natsuki Takaya
Her manga series "Fruits Basket", which debuted in 1998, became one of the best selling manga in North America. (Relevance Score: 2.34)
Charles Mason Hovey
Charles Mason Hovey was an American nurseryman, seed merchant, journalist and author of horticultural books best known for his two-volume large quarto, The Fruits of America published between 1848 and 1856. (Relevance Score: 2.41)
Fruits (book)
Fruits: A Caribbean Counting Poem (ISBN: 0805051716) is a children's picture book written by Valerie Bloom and illustrated by David Axtell. In 1997 it won the Nestlé Smarties Book Prize Bronze Award. (Relevance Score: 2.37)
LST: Charles Mason Hovey.
MLST: Natsuki Takaya.
CST: Document 1: The document states that Natsuki Takaya is the author of the manga series "Fruits Basket," which first appeared in 1998 and has gained significant popularity in North America. This information directly answers the query about the author of "Fruits." So, the document is relevant to the question.
Document 2: The document states that Charles Mason Hovey is the author of "The Fruits of America," a notable work published between 1848 and 1856. This directly answers the query regarding the author of "Fruits," confirming Hovey's authorship of a significant book on horticulture. So, the document is relevant to the question.
Document 3: The document identifies Valerie Bloom as the author of the children's picture book "Fruits: A Caribbean Counting Poem," which won the Nestlé Smarties Book Prize Bronze Award in 1997. This directly answers the query about the author of "Fruits," confirming Bloom's authorship. So, the document is relevant to the question.
Final Answer: Based on the analysis of the documents, the answer is Natsuki Takaya.
BC: Document 1: The document provides the name of the author, Natsuki Takaya, and mentions her work "Fruits Basket," which is a successful manga series. However, it does not clarify if "Fruits" refers to "Fruits Basket" or another work, leaving the specific question unanswered. So, the document is irrelevant to the question.
Document 2: The document identifies Charles Mason Hovey as an American nurseryman and the author of "The Fruits of America," a horticultural book published in the mid-19th century. However, it does not specify whether "Fruits" in the query refers to Hovey's work or another title, leaving the question of authorship unclear. So, the document is irrelevant to the question.
Document 3: The document states that "Fruits: A Caribbean Counting Poem" is a children's picture book written by Valerie Bloom and illustrated by David Axtell. It highlights that the book won the Nestlé Smarties Book Prize Bronze Award in 1997. This directly answers the query regarding the author of "Fruits," confirming that Valerie Bloom is the author. So, the document is relevant to the question.
Final Answer: Based on the analysis of the documents, the answer is Valerie Bloom.

Table 2: Different models' behaviors for the same context. orange, green, red, and blue text are query, relevant document, irrelevant document and final output.

|     | Pop QA | Trivia QA | Web Questions | SQuAD |
| --- | --- | --- | --- | --- |
| LST | 2634 | 17120 | 5114 | 25910 |
| BC  | 776  | 5024  | 1360 | 7392  |

Table 3: The time efficiency (in seconds) of different methods.

second document, while MLST is misled by the first document, leading errors in the final results in both documents. CST incorrectly judges the first document, which negatively impacts the criteria for later decisions. In contrast, Backtracking Correction effectively learns how to handle similar situations during training and ultimately answers the question correctly. More cases can be found in Appendix B.

Figure 3 illustrates the final performance of BC compared with others. Both CST and BC outperform LST and MLST across all benchmarks, demonstrating that **learning reasoning chains is beneficial for enhancing the LLM's ability to assess documents.** Chain-of-Thought $C$ can be viewed as an

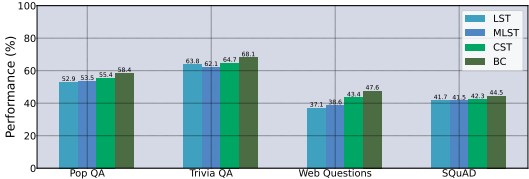

Figure 3: The performance of Backtracking Correction compared with other training methods.

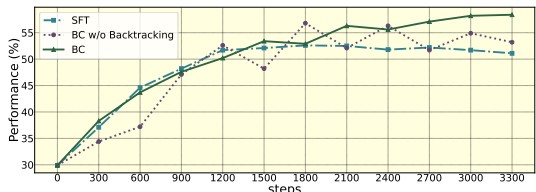

Figure 4: The training process of different methods on Pop QA.

additional supervision signal providing more information for learning alongside the label $L$. MLST does not consistently outperform LST. **However, the increased number of documents may improve the retrieval of relevant ones, which can also introduce additional noise.** Furthermore, BC achieves higher accuracy than CST, probably due to its more effective utilization of limited data and annotations. The details about the accuracy of classification results are listed in Appendix A.

## 4.4 EFFICIENCY ANALYSIS

Table 3 reveals the time efficiency for inference across four datasets, based on an A800 80GB system. The inference stage of our method is similar to CST, where all retrieved documents for a given query are analyzed within the same context. In contrast, LST processes each retrieved document separately. This results in a significant increase in the number of items to be processed, which limits the ability to fully leverage frameworks like vllm for speed optimization, despite having a shorter average context length.

## 4.5 ABLATION STUDIES

Figure 4 illustrates the training fluctuations of different settings on Pop QA. SFT exhibits better stability but a lower ceiling, as good generalization requires extensive annotated data. While improvements are substantial during the early steps, they become insignificant later on. In contrast, RL-based methods show greater fluctuations but achieve better overall performance. The backtracking algorithm simplifies the RL process by allowing optimization at each step to focus solely on the errors of the current state, as the remaining parts of the reasoning chains have already been learned. This approach enables BC to achieve a more stable training process.

## 5 CONCLUSION

This paper highlights the limitations of Label-Supervised Tuning (LST) and Chain-Supervised Tuning (CST) in retrieval-augmented generation (RAG). To address these issues, we reframe the task as a multi-step decision-making process and propose Backtracking Correction to train retrieval-augmented language models. Compared to existing strong baselines, our method achieves competitive performance without relying on additional data or annotations. Moreover, our approach is not limited to enhancing LLMs' reasoning abilities in RAG. We anticipate that it can be applied to other reasoning tasks like math and code in the future. However, training LLMs' correction abilities may prove to be more challenging outside the RAG context.

ACKNOWLEDGEMENT

The work described in this paper was partially funded by the National Natural Science Foundation of China (Grant No. 62272173), the Natural Science Foundation of Guangdong Province (Grant Nos. 2024A1515010089, 2022A1515010179), and the Science and Technology Planning Project of Guangdong Province (Grant No. 2023A0505050106).

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

|     | T | Pop QA | Trivia QA | Web Questions | SQuAD |
|-----|---|--------|-----------|---------------|-------|
| LST | 2 | 56.4 | 60.9 | 42.9 | 36.8 |
|     | 3 | 57.0 | 63.1 | 43.2 | 37.5 |
|     | 4 | 57.0 | 65.3 | 44.5 | 38.3 |
|     | 5 | 56.9 | 64.5 | 45.2 | 38.3 |
| CST | 2 | 58.1 | 66.8 | 46.2 | 44.0 |
|     | 3 | 58.8 | 67.3 | 46.6 | 44.8 |
|     | 4 | 59.2 | 67.3 | 47.1 | 45.4 |
|     | 5 | 58.6 | 67.0 | 47.7 | 45.2 |
| BC  | 2 | 60.9 | 70.4 | 50.6 | 46.8 |
|     | 3 | 61.7 | 71.2 | 51.4 | 47.3 |
|     | 4 | 62.0 | 71.5 | 51.5 | 47.6 |
|     | 5 | 61.7 | 71.6 | 51.5 | 47.3 |

Table 4: The hit rate of different methods under different $T$.

## A  THE ANALYSIS OF INTERMEDIATE RESULT

LST methods select the document with the highest score as the final context, making it hard to use accuracy to assess their classification abilities. For example, there may be more than one documents relevant to the given query. Only the one with the highest score is determined as 'relevant' and the others are all determined as 'irrelevant', However, the accuracy can not reflect models' classification abilities. Considering that, we adopt the hit rate of the documents determined as 'relevant'. The performance under different $T$ is listed in Table 4. The results show that BC achieves higher classification accuracy when judging the relevance of retrieved documents.

## B  CASE STUDY

We evaluate the performance of Backtracking Correction in extreme scenarios. There are no instances where all the retrieved documents are "relevant", but there are cases where all the retrieved documents are judged as "irrelevant". Table 5 shows an example. The model refuses to provide any answers, resulting in an accuracy of 0. Although there are a few similar samples in the training data, the model does not typically follow this pattern. We suspect this is due to the small proportion of such samples in the training set.

Besides, we also found some documents that have ambiguous relevance to the query. As shown in Table 6, multiple documents (Document 1, Document 2, Document 3, and Document 5) mention Henry Feilden (due to multiple namesakes). The gold answer in the dataset is "politician," indicating that Document 1 is relevant, while the others are not. However, it can be ambiguous to classify Documents 2, 3, and 5 as irrelevant, since they also mention the name "Henry Feilden." Such ambiguities are difficult to avoid during both training and inference, as the original human annotators may not have reviewed all documents with potentially ambiguous relevance to the query. As with previous work, we consider only those documents containing the gold answers as relevant.

What sport does Best play?
Best Player
Best Player is a 2011 television film that aired on Nickelodeon on March 12, 2011. The movie stars Jerry Trainor and Jennette McCurdy, from the show iCarly. Filming started on October 24, 2009 in Victoria, British Columbia, Canada and wrapped up production on November 18, 2009.
George Best
Best played for three clubs in the United States: Los Angeles Aztecs, Fort Lauderdale Strikers and later San Jose Earthquakes; he also played for the Detroit Express on a European tour. Best was a success on the field, scoring 15 goals in 24 games in his first season with the Aztecs and named as the NASL's best midfielder in his second. He and manager Ken Adam opened "Bestie's Beach Club" (now called "The Underground" after the London subway system) in Hermosa Beach, California in the 1970s, and continued to operate it until the 1990s.
Ultimate in Japan
Ultimate, originally called ultimate frisbee, is a non-contact team field sport played with a flying disc, invented in New Jersey, USA, in 1968. Japanese players and teams rose to prominence in the 1990s, and today are among the strongest competitors in the sport globally.
Football
Thus, the best association football and American football players are among the highest paid athletes in the world. Australian rules football has the highest spectator attendance of all sports in Australia. Similarly, Gaelic football is the most popular sport in Ireland in terms of match attendance, and the All-Ireland Football Final is the most watched event of that nation's sporting year. Rugby union is the most popular sport in New Zealand, Samoa, Tonga, and Fiji. It is also the fastest growing sport in the U.S. with college rugby being the fastest growing college sport in that country.
List of rural sports and games
Aunt Sally – An Oxfordshire game, it is the under arm throwing of the dolly (a truncheon shaped stick) at a suspended target. Each player in the team has 6 throws. The best score out of 24 wins. ; Bat and trap – An English bat-and-ball game, played in pub gardens mostly in Kent. ; Bog snorkelling – Competitors must complete two consecutive lengths of a water filled trench cut through a peat bog, in the shortest time possible. ; Coconut shy – Each player has 6 balls to throw at targets of coconuts balanced on raised stands.
BC: Document 1: The document is about a television film called "Best Player," which stars Jerry Trainor and Jennette McCurdy, but it doesn't mention a sport. It appears to be a film about a character named Best, possibly a nickname or a character's name, but the document does not specify what sport the character is associated with. So, the document is not relevant to the question.
Document 2: The document is about George Best, a former professional soccer player, and his career in the United States and his business ventures, specifically his nightclub, "Bestie's Beach Club." It mentions his playing career in the North American Soccer League (NASL) and his achievements, but does not mention any information about the sport he currently plays. The document is irrelevant to the question "What sport does Best play" because it does not provide any information about Best's current sport or activity, but rather focuses on his past career and personal life.
Document 3: The document is a brief overview of the sport of Ultimate, specifically its origins, its rise in popularity in Japan, and its global competitiveness. It is irrelevant to the question "What sport does Best play" because the document does not mention a person named Best. It is actually talking about the sport Ultimate, which is a team sport, not a person's name. So, the document is not relevant to the question.
Document 4: The document discusses the popularity and revenue of various types of football, including association football, American football, Australian rules football, Gaelic football, and Rugby union. It mentions that these sports have high revenue, high spectator attendance, and are popular in different countries. This document is irrelevant to the question "What sport does Best play" because it does not mention any specific individual or team named "Best" and does not provide information about their sport of choice. The document is a general overview of the popularity of different sports, but it does not contain any information that would help answer the question. So, the document is irrelevant to the question.
Document 5: The document appears to be a list of various traditional rural sports and games, including Aunt Sally, Bat and trap, Bog snorkelling, Coconut shy, and Cooper's Throw. This document is irrelevant to the question "What sport does Best play" because the document does not mention the name "Best" or any information about a person named Best participating in any of the listed sports. The question seems to be asking about an individual's sport, whereas the document lists various traditional sports and games in general. So, the document is irrelevant to the question.
Final Answer: The sport that Best plays is not uniquely determined by the provided documents.

Table 5: The case that Backtracking Correction handle an extreme case where all retrieved documents are entirely irrelevant.

What is Henry Feilden's occupation?

Henry Feilden (Conservative politician)

Henry Master Feilden (21 February 1818 Ž013 5 September 1875) was an English Conservative Party politician.

Feilden

Bernard Feilden (1919Ž0132008), British conservation architect ; Bob Feilden (1917Ž0132004), British mechanical engineer ; Gerry Feilden (1904Ž0131981), British general and horse racing identity ; Henry Feilden (disambiguation) ; Joseph Feilden (1824Ž0131895), British politician ; Richard Feilden (1950Ž0132005), British architect ; William Feilden (1772Ž0131850), British politician.

Feilden baronets

Henry Wemyss Feilden, second son of the second Baronet, was an Arctic explorer.

Bernard Feilden

Feilden was born in Hampstead, London. He was educated at Bedford School and The Bartlett School of Architecture, University College, London, completing his training at the Architectural Association after the second world war. His love of architecture was inherited from his grandfather, Brightwen Binyon (1846-1905), an Ipswich architect and former pupil of Alfred Waterhouse. He joined the practice of Edward Boardman and Son in Norwich, where he designed the Trinity United Reformed Church. He set up an architectural practice, Feilden+Mawson, with David Mawson in 1956, to which offices in Norwich, London and Cambridge were later added.

Henry Wemyss Feilden

Colonel Henry Wemyss Feilden, CB (6 October 1838 Ž013 8 June 1921) was a British Army officer, Arctic explorer and naturalist.

BC: Document 1: The document is a brief biography of Henry Feilden, stating that he was an English Conservative Party politician. It is relevant to the question about Henry Feilden's occupation because it explicitly mentions that he was a politician, specifically a Conservative politician, which directly answers the question. So, the document is relevant to the question.

Document 2: The document lists various individuals with the surname Feilden, including their birth and death years, along with a brief description of their profession or notable achievements.Henry Feilden is mentioned as a disambiguation, but no specific information is provided about his occupation. However, upon examining the list, there is no information about a notable Henry Feilden who is a well-known figure, suggesting that the document is likely referring to the disambiguation page, which is a placeholder for a person with that name. So, the document is relevant to the question, but the answer cannot be definitively determined from the provided information.

Document 3: The document briefly describes Henry Feilden as the second son of the second Baronet and mentions that he was an Arctic explorer. This document is relevant to the question because it provides information about Henry Feilden's occupation, specifically that he was an Arctic explorer.

Document 4: The document briefly describes the life and career of Bernard Feilden, an architect, including his education, family background, and professional accomplishments. It mentions his love for architecture, his apprenticeship with Edward Boardman and Son, and his establishment of his own architectural practice, Feilden+Mawson. This document is irrelevant to the question "What is Henry Feilden's occupation" because the document does not mention a person named Henry Feilden, but rather Bernard Feilden, and there is no information about Henry's occupation.

Document 5: The document is a brief biographical description of Henry Feilden, stating that he was a British Army officer, Arctic explorer, and naturalist. This information is relevant to the question "What is Henry Feilden's occupation?" because it directly answers the question by listing his occupations, which include being a British Army officer, an Arctic explorer, and a naturalist. So, the document is relevant to the question.

Final Answer: Based on the analysis of the documents, the answer is Arctic explorer.

Table 6: The case that Backtracking Correction handle an extreme case where all retrieved documents are entirely irrelevant.

