# OpenReview forum: "Training Large Language Models for Retrieval-Augmented Question Answering through Backtracking Correction"
_ICLR.cc/2025/Conference — ICLR 2025 Poster_

### Official Review · Reviewer_c4sg · 2024-10-31

**Soundness:** 3
**Presentation:** 3
**Contribution:** 3
**Rating:** 6
**Confidence:** 3

**Summary:**

The paper introduces Backtracking Correction, a novel method to enhance RAG by transforming it into a multi-step decision-making process. BC optimizes models through error sampling and self-correction, starting from the final step and backtracking to previous states, simplifying the learning of reasoning chains and improving robustness against noisy documents.

**Strengths:**

1.BC's backtracking algorithm provides a fresh perspective on training LLMs for RAG, simplifying complex reasoning into manageable steps, which is innovative in the context of handling noisy information in document retrieval.
2.By eliminating the need for extensive annotations, BC lowers the barrier for training complex LLMs, making it more accessible and cost-effective compared to methods relying on detailed reasoning chain annotations.
3.The paper demonstrates BC's effectiveness through extensive experiments, showing improved performance over strong baselines, including fine-tuned LLMs and proprietary models, highlighting the practical value of the proposed method.

**Weaknesses:**

1.The paper lacks an adequate exploration of BC's limitations in processing larger datasets and executing more complex reasoning tasks, like multi-hop reasoning. Including a thorough discussion on these aspects could significantly strengthen the analysis.
2.While BC is proposed as a simplified approach, the backtracking process might still be computationally expensive, especially for environments with limited resources, which could hinder its adoption.
* The numbering of the equations contains errors, which has resulted in confusion in their citations throughout the paper.

**Questions:**

1.Could the authors share additional insights into the error analysis of the model, particularly in the context of multi-step reasoning questions? A breakdown of errors by reasoning step or a comparison of error types between BC and baseline methods would be meaningful.
2.What impact does an increase in the number of documents have on BC's performance? It would be beneficial to see experiments demonstrating how performance metrics change as the number of retrieved documents increases.
3.Can Backtracking Correction effectively handle extreme cases where all retrieved documents are either highly relevant or entirely irrelevant? Evaluating BC's performance in these scenarios could provide valuable insights.
4.How does Backtracking Correction manage documents that have ambiguous relevance to the query? Furthermore, what effect does this ambiguity have on the model's final output? A detailed analysis of these cases would enhance the understanding of BC's robustness.

---

> ### Author Response · Authors · 2024-11-18
> **Response to Reviewer c4sg (part 1)**
>
> Thanks for your advice. We will respond to your questions point by point.
>
> 1. We chose to work with current tasks rather than more complex reasoning tasks (e.g., math reasoning, multi-hop RAG) for the following reasons:
>
> (1) **Short answers suitable for pattern matching**: These tasks typically involve short answers that can be directly evaluated through pattern matching. In contrast, more complex RAG-QA tasks often rely on closed-source models like GPT-4 to assess answers (e.g., Arena). We believe that training on data annotated by the models themselves is crucial for achieving good performance, as these evaluators tend to prefer answers that align with their own distribution. Unfortunately, due to limited access to GPT, it is challenging to conduct large-scale annotations and evaluations.
>
> (2) **Ease of error identification and correction**: It is relatively easy to locate and correct errors in the reasoning chains for these tasks. Specifically, we can directly identify reasoning steps where the model’s judgments (relevant or irrelevant) deviate from the gold labels. The gold labels are included in the original dataset, making error identification straightforward. Additionally, we can leverage the model’s ability to summarize consistent and conflicting points (a skill that is easy to learn) to correct incorrect judgments. In contrast, for more complex tasks like math reasoning, it is difficult to locate and correct errors within the context without the assistance of strong closed-source models. Addressing these challenges in a cost-effective way will be part of our future work.
>
> We will discuss this further in the Limitations section.
>
> 2. The annotation process for training data is indeed the most resource-intensive part. Previous work (LST and CST) relies on powerful closed-source models to annotate the data, but API calls to these proprietary LLMs are both time-consuming and costly. In contrast, our method uses the target model itself to generate pairwise responses, which can be accelerated using frameworks like vllm.
> Regarding GPU resources for training, context length plays a key role in resource demands. Self-RAG (LST) requires 9 hours on 8 * A800 80GB GPUs, whereas our method takes 14 hours, as it requires a longer context length. While other methods do not publish their training datasets, the pattern of Chain-of-Notes (CST) data is likely similar to ours, where all retrieved documents for each query are analyzed within the same context.

---

> ### Author Response · Authors · 2024-11-18
> **Response to Reviewer c4sg (part 2)**
>
> 3. Here we show an example:
>
> Query: What are forbidden on michigan's mackinac island?
>
> Document: [Mackinac Island] Bicycles, roller skates/roller blades, carriages, and saddle horses are available for rent. An road follows the island's perimeter, and numerous roads, trails and paths cover the interior. M-185, the United States' only state highway without motorized vehicles, makes a circular loop around the island, closely hugging the shoreline. The island is the location of Mackinac Island State Park, which covers approximately 80 percent of the island and includes Fort Mackinac as well as portions of the island's historic downtown and harbor. No camping is allowed on the island, but numerous hotels and bed.
>
> Original Judge: According to the document, there is no explicit mention of what is forbidden on Mackinac Island. However, the document does mention that ‘No camping is allowed on the island’ which implies that camping is forbidden. This document is relevant to the question because it provides information about the rules and regulations on Mackinac Island, specifically mentioning what is not allowed, which is camping. So, the document is relevant to the question.
>
> Corrected Judge: Summary of the document: The document provides information about Mackinac Island, including its transportation options, road network, and attractions such as Mackinac Island State Park and Fort Mackinac. It also mentions that camping is not allowed on the island, but there are numerous hotels and bed and breakfast options available. The document is irrelevant to the question ‘What are forbidden on Michigan's Mackinac Island’ because it mentions what is allowed on the island, such as renting bicycles, roller skates/roller blades, carriages, and saddle horses, and what is not allowed (camping), but does not explicitly state what is forbidden. The document's focus is on the island's attractions, transportation, and accommodations, rather than its restrictions. So, the document is irrelevant to the question.
>
> As shown in the example, the model misjudged the document at first. However, provided with the prompt in the pink box on page 7 with the label ‘irrelevant’ (which has been already involved in the dataset), the model finds a conflicting point – the document does not explicitly state what is forbidden. In this way, we can utilize the model’s summarization ability to correct the wrong judgement. By contrast, the previous work does not consider the mistakes made by the target model. They utilize closed-source models to construct the training data, remove those containing errors, and conduct training on the filtered data.
>
> We will add more examples in our appendix.
>
> 4. The hit rate of the documents determined as ‘relevant’ for different numbers of documents are listed below:
>
> Self-RAG (LST)
>
> |  | Pop QA | Trivia QA | Web Questions | SQuAD |
> | :----: | :----: | :----: | :----: | :----: |
> | T=2 | 56.4 | 60.9 | 42.9 | 36.8 |
> | T=3 | 57.0 | 63.1 | 43.2 | 37.5 |
> | T=4 | 57.0 | 65.3 | 44.5 | 38.3 |
> | T=5 | 56.9 | 64.5 | 45.2 | 38.3 |
>
> CST (Chain-of-Notes does not publish their data so we use llama-3-8b-instruct to annotate the data)
>
> |  | Pop QA | Trivia QA | Web Questions | SQuAD |
> | :----: | :----: | :----: | :----: | :----: |
> | T=2 | 58.1 | 66.8 | 46.2 | 44.0 |
> | T=3 | 58.8 | 67.3 | 46.6 | 44.8 |
> | T=4 | 59.2 | 67.3 | 47.1 | 45.4 |
> | T=5 | 58.6 | 67.0 | 47.7 | 45.2 |
>
> BC (Ours)
>
> |  | Pop QA | Trivia QA | Web Questions | SQuAD |
> | :----: | :----: | :----: | :----: | :----: |
> | T=2 | 60.9 | 70.4 | 50.6 | 46.8 |
> | T=3 | 61.7 | 71.2 | 51.4 | 47.3 |
> | T=4 | 62.0 | 71.5 | 51.5 | 47.6 |
> | T=5 | 61.7 | 71.6 | 51.5 | 47.3 |
>
> LST methods select the document with the highest score as the context, making them unsuitable for T = 1. The results show that BC achieves higher classification accuracy when judging the relevance of the retrieved documents. Additionally, as the number of retrieved documents increases, the model initially benefits from the higher hit rate of relevant documents but eventually becomes distracted by the growing noise introduced by irrelevant documents. We will include this analysis in our appendix.

---

> ### Author Response · Authors · 2024-11-18
> **Response to Reviewer c4sg (part 3)**
>
> 5. There are no instances where all the retrieved documents are relevant, but there are cases where all the retrieved documents are judged as "irrelevant." Here’s an example:
>
> Q: What sport does Best play?
>
> Document1: [Best Player]
> Best Player is a 2011 television film that aired on Nickelodeon on March 12, 2011. The movie stars Jerry Trainor and Jennette McCurdy, from the show iCarly. Filming started on October 24, 2009 in Victoria, British Columbia, Canada and wrapped up production on November 18, 2009.
>
> Document2: [George Best]
> Best played for three clubs in the United States: Los Angeles Aztecs, Fort Lauderdale Strikers and later San Jose Earthquakes; he also played for the Detroit Express on a European tour. Best was a success on the field, scoring 15 goals in 24 games in his first season with the Aztecs and named as the NASL's best midfielder in his second. He and manager Ken Adam opened "Bestie's Beach Club" (now called "The Underground" after the London subway system) in Hermosa Beach, California in the 1970s, and continued to operate it until the 1990s.
>
> Document3: [Ultimate in Japan]
> Ultimate, originally called ultimate frisbee, is a non-contact team field sport played with a flying disc, invented in New Jersey, USA, in 1968. Japanese players and teams rose to prominence in the 1990s, and today are among the strongest competitors in the sport globally.
>
> Document4: [Football]
> Thus, the best association football and American football players are among the highest paid athletes in the world. Australian rules football has the highest spectator attendance of all sports in Australia. Similarly, Gaelic football is the most popular sport in Ireland in terms of match attendance, and the All-Ireland Football Final is the most watched event of that nation's sporting year. Rugby union is the most popular sport in New Zealand, Samoa, Tonga, and Fiji. It is also the fastest growing sport in the U.S. with college rugby being the fastest growing college sport in that country.
>
> Document5: [List of rural sports and games]
> Aunt Sally – An Oxfordshire game, it is the under arm throwing of the dolly (a truncheon shaped stick) at a suspended target. Each player in the team has 6 throws. The best score out of 24 wins. ; Bat and trap – An English bat-and-ball game, played in pub gardens mostly in Kent. ; Bog snorkelling – Competitors must complete two consecutive lengths of a water filled trench cut through a peat bog, in the shortest time possible. ; Coconut shy – Each player has 6 balls to throw at targets of coconuts balanced on raised stands.
>
> Response:
>
> Here are the analyses for each document:
>
> Document1: The document is about a television film called "Best Player," which stars Jerry Trainor and Jennette McCurdy, but it doesn't mention a sport. It appears to be a film about a character named Best, possibly a nickname or a character's name, but the document does not specify what sport the character is associated with.
>
> So, the document is not relevant to the question.
>
> Document2: The document is about George Best, a former professional soccer player, and his career in the United States and his business ventures, specifically his nightclub, "Bestie's Beach Club." It mentions his playing career in the North American Soccer League (NASL) and his achievements, but does not mention any information about the sport he currently plays.
>
> The document is irrelevant to the question "What sport does Best play" because it does not provide any information about Best's current sport or activity, but rather focuses on his past career and personal life.
>
> Document3: The document is a brief overview of the sport of Ultimate, specifically its origins, its rise in popularity in Japan, and its global competitiveness.
>
> It is relevant to the question "What sport does Best play" because the document does not mention a person named Best. It is actually talking about the sport Ultimate, which is a team sport, not a person's name. So, the document is not relevant to the question.
>
> Document4: The document discusses the popularity and revenue of various types of football, including association football, American football, Australian rules football, Gaelic football, and Rugby union. It mentions that these sports have high revenue, high spectator attendance, and are popular in different countries.
>
> This document is irrelevant to the question "What sport does Best play" because it does not mention any specific individual or team named "Best" and does not provide information about their sport of choice. The document is a general overview of the popularity of different sports, but it does not contain any information that would help answer the question. So, the document is irrelevant to the question.

---

> ### Author Response · Authors · 2024-11-18
> **Response to Reviewer c4sg (part 4)**
>
> Document5: The document appears to be a list of various traditional rural sports and games, including Aunt Sally, Bat and trap, Bog snorkelling, Coconut shy, and Cooper's Throw.
>
> This document is irrelevant to the question "What sport does Best play" because the document does not mention the name "Best" or any information about a person named Best participating in any of the listed sports. The question seems to be asking about an individual's sport, whereas the document lists various traditional sports and games in general. So, the document is irrelevant to the question.
>
> Final Answer: The sport that Best plays is not uniquely determined by the provided documents.
>
> The model **refuses to provide any answers**, resulting in an accuracy of 0. Although there are a few similar samples in the training data, the model does not typically follow this pattern. We suspect this is due to the small proportion of such samples in the training set.
>
> 6. It is a really valuable question. We found some documents that have ambiguous relevance to the query. Here’s an example:
>
> Q: What is Henry Feilden's occupation?
>
> Document1: [Henry Feilden (Conservative politician)]
> Henry Master Feilden (21 February 1818 \u2013 5 September 1875) was an English Conservative Party politician.
>
> Document2: [Feilden]
> Bernard Feilden (1919\u20132008), British conservation architect ; Bob Feilden (1917\u20132004), British mechanical engineer ; Gerry Feilden (1904\u20131981), British general and horse racing identity ; Henry Feilden (disambiguation) ; Joseph Feilden (1824\u20131895), British politician ; Richard Feilden (1950\u20132005), British architect ; William Feilden (1772\u20131850), British politician.
>
> Document3: [Feilden baronets]
> Henry Wemyss Feilden, second son of the second Baronet, was an Arctic explorer.
>
> Document4: [Bernard Feilden]
> Feilden was born in Hampstead, London. He was educated at Bedford School and The Bartlett School of Architecture, University College, London, completing his training at the Architectural Association after the second world war. His love of architecture was inherited from his grandfather, Brightwen Binyon (1846-1905), an Ipswich architect and former pupil of Alfred Waterhouse. He joined the practice of Edward Boardman and Son in Norwich, where he designed the Trinity United Reformed Church. He set up an architectural practice, Feilden+Mawson, with David Mawson in 1956, to which offices in Norwich, London and Cambridge were later added.
>
> Document5: [Henry Wemyss Feilden]
> Colonel Henry Wemyss Feilden, CB (6 October 1838 \u2013 8 June 1921) was a British Army officer, Arctic explorer and naturalist.
>
> Response:
>
> Here are the analyses for each document:
>
> Document1: The document is a brief biography of Henry Feilden, stating that he was an English Conservative Party politician. It is relevant to the question about Henry Feilden's occupation because it explicitly mentions that he was a politician, specifically a Conservative politician, which directly answers the question.
>
> So, the document is relevant to the question.
>
> Document2: The document lists various individuals with the surname Feilden, including their birth and death years, along with a brief description of their profession or notable achievements.Henry Feilden is mentioned as a disambiguation, but no specific information is provided about his occupation. However, upon examining the list, there is no information about a notable Henry Feilden who is a well-known figure, suggesting that the document is likely referring to the disambiguation page, which is a placeholder for a person with that name.
>
> So, the document is relevant to the question, but the answer cannot be definitively determined from the provided information.
>
> Document3: The document briefly describes Henry Feilden as the second son of the second Baronet and mentions that he was an Arctic explorer.
>
> This document is relevant to the question because it provides information about Henry Feilden's occupation, specifically that he was an Arctic explorer.
>
> Document4: The document briefly describes the life and career of Bernard Feilden, an architect, including his education, family background, and professional accomplishments. It mentions his love for architecture, his apprenticeship with Edward Boardman and Son, and his establishment of his own architectural practice, Feilden+Mawson.
>
> This document is irrelevant to the question "What is Henry Feilden's occupation" because the document does not mention a person named Henry Feilden, but rather Bernard Feilden, and there is no information about Henry's occupation.

---

> ### Author Response · Authors · 2024-11-18
> **Response to Reviewer c4sg (part 5)**
>
> Document5: The document is a brief biographical description of Henry Feilden, stating that he was a British Army officer, Arctic explorer, and naturalist. This information is relevant to the question "What is Henry Feilden's occupation?" because it directly answers the question by listing his occupations, which include being a British Army officer, an Arctic explorer, and a naturalist.
>
> So, the document is relevant to the question.
>
> Final Answer: Based on the analysis of the documents, the answer is Arctic explorer.
>
> Multiple documents (Document 1, Document 2, Document 3, and Document 5) mention Henry Feilden (due to multiple namesakes). The gold answer in the dataset is "politician," indicating that Document 1 is relevant, while the others are not. However, it can be ambiguous to classify Documents 2, 3, and 5 as irrelevant, since they also mention the name "Henry Feilden." Such ambiguities are difficult to avoid during both training and inference, as the original human annotators may not have reviewed all documents with potentially ambiguous relevance to the query. As with previous work, we consider only those documents containing the gold answers as relevant.

---

> > ### Comment · Reviewer_c4sg · 2024-11-28
> >
> > Thank you for the experiments and case analysis you provided, which addressed some of my concerns. I believe it would be beneficial to include this content in the appendix, and I will consider increasing the score as a result.

---

### Official Review · Reviewer_ED7n · 2024-11-03

**Soundness:** 3
**Presentation:** 3
**Contribution:** 3
**Rating:** 6
**Confidence:** 5

**Summary:**

This paper focuses on issues with RAG models having difficulties when encountering irrelevant documents and in particular, in sequential decision making processes where documents are classified as relevant or irrelevant before a final answer is produced. The authors explain issues with existing methods for chain and label based supervised fine tuning and instead propose a backtracking correction method. Overall the paper is well written and, for the most part well presented.

While the proposed method shows some modest gains, Its hard to determine the full benefits as there is only one comparison of the same model being trained with backtracking vs baselines. It would be informative to see more comparisons of models in the `Fine-tuned w/retrieval `section of table 1 also evaluated using backtracking.  Additionally, it would be useful to see a bit more detail on the experimental setup and some analysis or detail on how much more expensive the backtracking method is to use.

**Strengths:**

- Overall, the paper is well written, organized, and presented with informative figures
- the presented method demonstrates some gains over baseline methods

**Weaknesses:**

- only a single model is shown comparing baseline training methods against the proposed method with modest gains
- evaluation datasets are all fairly old and most certainly appear in the training data of many of the investigated models
- the presentation of sec 3.3 is overly complex and may benefit from some editing, subsections, or the addition of a simple figure

**Questions:**

- what is the wallclock train time of the different methods?
- does the method improve over baseline training setups for other models? Only llama2 7B is shown in fig 3 but llama3-8b was also trained with backtracking as could many of those found in fine-tuned w/ retrieval section of table 1
- what is the accuracy of relevance /irrelevance classification with different methods? afaict only end-to-end answering performance is shown

---

> ### Author Response · Authors · 2024-11-18
> **Response to Reviewer ED7n**
>
> Thanks for your advice. We will respond to your questions point by point.
>
> 1. We begin with **Llama-2-7B-base**, as used in previous work (e.g., Self-RAG, REAR, Chain-of-Notes), to ensure a fair comparison (see Section 4.1.1). After initializing the model’s meta-skills, we can perform Backtracking Correction without requiring additional data or annotations, while still achieving competitive performance compared to methods that rely on proprietary LLM annotations. The main contribution of our approach is the ability to eliminate the need for external annotations. We also conducted experiments using **Llama-3-8B-base** to demonstrate the generalizability of our method, but currently, there are no available baselines using the same backbone for comparison. We use **base models**, rather than **chat models**, as backbones to ensure that the fundamental capabilities of the models have minimal influence on their final performance.
>
> 2. The results show that most chat models have relevant knowledge for Trivia QA and Web Questions. However, Pop QA is relatively new, and nearly all LLMs struggle to answer questions on this dataset directly. We selected these datasets primarily for their convenience in evaluation and error correction. The model's ability to summarize consistent or conflicting points allows us to correct the incorrect steps, while the final answers can be directly evaluated through pattern matching.
>
> Recent works have introduced more complex datasets (e.g., math reasoning, multi-hop RAG, etc.), but locating and correcting errors in these tasks remains a challenging issue. Currently, this is difficult to achieve without the assistance of powerful closed-source models. In future work, we aim to apply our method to such tasks.
>
> 3. We will divide it into several subsections.
>
> 4. Annotation of training data is indeed the most time-consuming process. Previous work (LST and CST) relies on powerful closed-source models to annotate the data, but API calls to these proprietary LLMs are both time-intensive and costly. In contrast, our method leverages the target model itself to generate pairwise responses, which can be accelerated using frameworks like vllm.
> Regarding actual GPU training hours, the context length plays a significant role in time efficiency. For example, Self-RAG (LST) takes 9 hours using 8 * A800 80GB GPUs, while our method takes 14 hours, as it requires a longer context length. Although other methods have not published their training datasets, the pattern of Chain-of-Notes (CST) data is likely similar to ours, where all retrieved documents for each query are analyzed within the same context.
>
> 5. The performance of the base models and fine-tuned models are listed below:
> |  | Pop QA | Trivia QA | Web Questions | SQuAD |
> | :----: | :----: | :----: | :----: | :----: |
> | LL2-7B-base | 4.5 | 10.9 | 6.1 | 3.7 |
> | LL3-8B-base | 5.7 | 10.8 | 4.4 | 3.8 |
> | LL2-7B-chat | 25.1 | 58.7| 48.6 | 19.1 |
> | LL3-8B-instruct | 25.5 | 64.09 | 50.89 | 20.69 |
> We will update the relevant results in tables.
>
> 6. LST methods select the document with the highest score as the final context, making it hard to use accuracy to assess their classification abilities. For example, there may be more than one documents relevant to the given query. Only the one with the highest score is determined as ‘relevant’ and the others are all determined as ‘irrelevant’, However, the accuracy can not reflect models’ classification abilities. Considering that, we adopt the hit rate of the documents determined as ‘relevant’. The performance under different $T$ is listed below:
>
> Self-RAG (LST)(LST methods select the document with the highest score as the context, making them unsuitable for T = 1).
>
> |  | Pop QA | Trivia QA | Web Questions | SQuAD |
> | :----: | :----: | :----: | :----: | :----: |
> | T=2 | 56.4 | 60.9 | 42.9 | 36.8 |
> | T=3 | 57.0 | 63.1 | 43.2 | 37.5 |
> | T=4 | 57.0 | 65.3 | 44.5 | 38.3 |
> | T=5 | 56.9 | 64.5 | 45.2 | 38.3 |
>
> CST (Chain-of-Notes does not publish their data so we use llama-3-8b-instruct to annotate the data)
>
> |  | Pop QA | Trivia QA | Web Questions | SQuAD |
> | :----: | :----: | :----: | :----: | :----: |
> | T=2 | 58.1 | 66.8 | 46.2 | 44.0 |
> | T=3 | 58.8 | 67.3 | 46.6 | 44.8 |
> | T=4 | 59.2 | 67.3 | 47.1 | 45.4 |
> | T=5 | 58.6 | 67.0 | 47.7 | 45.2 |
>
> BC (Ours)
>
> |  | Pop QA | Trivia QA | Web Questions | SQuAD |
> | :----: | :----: | :----: | :----: | :----: |
> | T=2 | 60.9 | 70.4 | 50.6 | 46.8 |
> | T=3 | 61.7 | 71.2 | 51.4 | 47.3 |
> | T=4 | 62.0 | 71.5 | 51.5 | 47.6 |
> | T=5 | 61.7 | 71.6 | 51.5 | 47.3 |
>
> The results show that BC achieves higher classification accuracy when judging the relevance of retrieved documents. We will include this analysis in the appendix.

---

### Official Review · Reviewer_hZRq · 2024-11-03

**Soundness:** 2
**Presentation:** 3
**Contribution:** 3
**Rating:** 6
**Confidence:** 3

**Summary:**

This paper proposes an interesting Backtracking Correction method to enhance LLMs' capability on RAG QA. Specifically, they employ Self-play algorithm to correct LLM's own CoT errors in a backward manner, and leverage the reward difference between the corrected and original outputs to train LLMs.  The authors present positive results by comparing their proposed algorithm comparing with Proprietary LLMs, SFT without retrieval, and previous SOTA SFT retrieval methods. An example and some analysis presented to help readers understand how their method work.

**Strengths:**

1. The idea of the paper is quite interesting in that it leverages the self-play algorithm to produce corrected CoT, which mitigates the lack of annotations problems.
2. The paper is well structured so it is relatively easy to follow despite the complexity (though it can be made clearer - see my comments below).
3. Results are in general in favor of the proposed method and some further analysis presented to help readers understand the improvements.

**Weaknesses:**

1. The overall structure of the paper is clear. However, some important details are missing.
- I think some notations are not clearly defined. For example, in Section 3.3, what exactly are $\Pi_{ref}$ and $o_t^-$?
- In Eq. 6, there appears to be inputs and outputs after the chain of equations, what are those? If this is an iterative process, I suggest using an actual example to illustrate better.
- Some math derivations are not immediately clear. For example, Eq. 10 and Eq. 12. I suggest the authors to add more details here or use the appendix to elaborate more.

2. Experiments can be more thorough.
- In Table 1 and Figure 3, we could conclude that this methods works for Llama-2-7b models, but does it work for other models? Adding experiments here will make the paper stronger.
- Similarly, most QA datasets are short-form factoid QA, would this method generalizes to other more complicated RAG-QA tasks, for example: https://arxiv.org/abs/2407.13998?

3. The authors show 1 example in Table 2, which is helpful, but I encourage them to dive deep by analyzing more examples and showing readers exactly how exactly correction of CoT works.

**Questions:**

- The naming of backtracking is a bit unintuitive. A typical backtracking algorithm starts with T=0, explore all path, and then backtrack. Here you don't have the initial explorations. Have you considered a less confusing name?
- Have you tried to shuffle the order of the documents and try your method there? How would that change the results?

---

> ### Author Response · Authors · 2024-11-18
> **Response to Reviewer hZRq (part 1)**
>
> Thanks for your advice. We will respond to your questions point by point.
>
> 1. Πref means the reference model. ot- is the previous model’s output, ot is the current model’s output , and ot+ is the next model’s output (corrected from ot). We will define them explicitly.
>
> 2. We will add some examples in the appendix to provide further clarification for better understanding. Please refer to the response to Reviewer gD7f and Reviewer c4sg.
>
> 3. The mathematical derivation for Eq. 10 (the optimal solution to Eq. 9) follows a similar approach to previous works ([1] and [2]). We will elaborate on this in the appendix for further clarity. As for Eq. 12, we move r(s, o) to the left side of the equation to derive it.
>
> [1] Jan Peters and Stefan Schaal. Reinforcement learning by reward-weighted regression for operational space control. In Zoubin Ghahramani (ed.), Machine Learning, Proceedings of the Twenty-Fourth International Conference (ICML 2007), Corvallis, Oregon, USA, June 20-24, 2007, volume 227 of ACM International Conference Proceeding Series, pp. 745–750. ACM, 2007. doi: 10.1145/1273496.1273590.
>
> [2] Xue Bin Peng, Aviral Kumar, Grace Zhang, and Sergey Levine. Advantage-weighted regression: Simple and scalable off-policy reinforcement learning. CoRR, abs/1910.00177, 2019.
>
> 4. We begin with Llama-2-7B-base, as used in previous work (e.g., Self-RAG, REAR, Chain-of-Notes), to ensure a fair comparison (see Section 4.1.1). To date, these works have not experimented with other backbones or published models. We use **base models**, rather than **chat models**, as backbones to ensure that the fundamental capabilities of the models have minimal influence on their final performance.
>
> We also experiment with Llama-3-8B-base to demonstrate the generalizability of our method; however, there are currently no baselines using the same backbone for comparison. Additionally, we have conducted some extra ablation experiments to strengthen the paper:
>
> The performance of the base models:
> |  | Pop QA | Trivia QA | Web Questions | SQuAD |
> | :----: | :----: | :----: | :----: | :----: |
> | LL2-7B-base | 4.5 | 10.9 | 6.1 | 3.7 |
> | LL3-8B-base | 5.7 | 10.8 | 4.4 | 3.8 |
>
> The performance of the chat models:
> |  | Pop QA | Trivia QA | Web Questions | SQuAD |
> | :----: | :----: | :----: | :----: | :----: |
> | LL2-7B-chat | 25.1 | 58.7| 48.6 | 19.1 |
> | LL3-8B-instruct | 25.5 | 64.09 | 50.89 | 20.69 |
>
> We will update the results in ablation study.
>
> 5. We choose tasks from short-form factoid QA for the following reasons:
>
> (1) Short answers suitable for pattern matching: These tasks typically involve short, direct answers that can be easily evaluated through pattern matching. In contrast, more complex RAG-QA tasks often rely on closed-source models like GPT-4 for answer evaluation (e.g., Arena). We believe that training on data annotated by the model itself is crucial for achieving strong performance, as these evaluators tend to favor answers that align with their own distribution. Unfortunately, due to limited GPT calls, it is challenging to conduct annotations and evaluations at scale.
>
> (2) Ease of error identification and correction: In short-form factoid QA, it is relatively easy to locate and correct errors in the reasoning chain. Specifically, we can directly identify reasoning steps where the model’s judgments (relevant or irrelevant) do not align with the gold labels. The gold labels are provided in the original dataset, making error localization straightforward. Additionally, we can leverage the model’s ability to summarize consistent and conflicting points (a skill that is easy to learn) to correct incorrect judgments. In contrast, for more complex tasks like mathematical reasoning, we believe it is difficult to identify and correct errors in the reasoning chain without the assistance of powerful closed-source models. Addressing these challenges cost-effectively will be the focus of our future work.
>
> 6. We will add more examples in our appendix. Please refer to the response to Reviewer gD7f and Reviewer c4sg.
>
> 7. Considering that the correction process involves backtracking from the leaves to the root, we have named the method "Backtracking Correction." Would "Backward Correction" be more appropriate?

---

> > ### Comment · Reviewer_hZRq · 2024-11-22
> > **Reply to the authors**
> >
> > Thanks for your responses!
> >
> > I think the justifications provided for the weakness are reasonable, but not sufficient to for a rating boost. So I will keep my original scores.

---

> ### Author Response · Authors · 2024-11-18
> **Response to Reviewer hZRq (part 2)**
>
> 8. Actually, this is what we did before training the model. We wanted to avoid any bias in training that might arise from the order in which the retriever ranks documents. Consider the scenario where we do not shuffle the documents. In this case, documents appearing earlier are more likely to be deemed "relevant." The model would learn this bias and become more inclined to classify the first few documents as "relevant." This would be undesirable, as we aim for the model to learn reasoning abilities rather than such ordering biases.
>
> Of course, BC introduces a dependency between the strictness or looseness of the standards used to assess the relevance of each retrieved document. Let’s consider the process of assessing three documents. In decoder-only architectures, if the model evaluates the first two documents with a loose standard, it is likely to classify the third document as "relevant," even if it is irrelevant, and vice versa. In other words, the standard used to assess the relevance of the current document is influenced by the standards applied to previous documents in the context. This is akin to the few-shot settings in in-context learning, which are determined by the decoder architecture.

---

### Official Review · Reviewer_ofNF · 2024-11-04

**Soundness:** 3
**Presentation:** 4
**Contribution:** 3
**Rating:** 6
**Confidence:** 4

**Summary:**

In the RAG setting, it is sometimes required to distinguish whether each document is relevant or irrelevant, and stepwise reasoning may be utilized for this purpose. However, training an LLM for this task poses challenges such as the need for extensive annotation. This paper proposes Backtracking Correction, which reformulates the stepwise reasoning of RAG as a decision process for each document. It separates the decision process of the multiple documents into context space (given the correct decision) and decision space (model output) starting from the final state, enabling an easy-to-hard progression.

**Strengths:**

- It is very well-written, and the figures are well-illustrated, making it easy to follow.
- A well-defined problem with a reasonable proposed method.

**Weaknesses:**

There are some concerns with the experimental setup.
- Fair comparison with baselines: Why weren’t the base models of Backtracking Correction (LL2-7B-base, LL3-8B-base) used as baselines? Non-chat base models could also have been fine-tuned. Including the base models in the baseline or incorporating chat models into Backtracking would ensure a fair comparison.
- "Without relying on additional data or annotations." Lines 411-416 mention that competitive performance can be achieved without additional training data (e.g., chat models). However, the model used to obtain the original annotation itself is a chat model. Is there a reason why the model used for SCS original annotation (line 326) and the backbone model were set differently?
- (Very minor) There is a typo "err" in the abstract, and it does not match the abstract on OpenReview.

**Questions:**

- Please refer to the mentioned weaknesses.
- Also, as the number of documents increases, the binary decision tree is likely to grow very rapidly. Is there a way to sample better paths without considering all paths in the decision space during training?

---

> ### Author Response · Authors · 2024-11-18
> **Response to Reviewer ofNF**
>
> Thanks for your advice. We will respond to your questions point by point.
> 1. To ensure a fair comparison, we begin with the **base models** (Llama-2-7B-base and Llama-3-8B-base), as used in previous works (e.g., Self-RAG, REAR, Chain-of-Notes) (Section 4.1.1). However, these base models exhibit **poor instruction-following performance**, often failing to match the expected output patterns, which makes error identification and sampling difficult. To address this, we use Llama-3-8B-Instruct to generate the original annotations for a small SFT dataset, S_{CS}, derived from the training data, which is then used to **initialize the base models**.
>
> The performance of the base models are listed below:
>
> |  | Pop QA | Trivia QA | Web Questions | SQuAD |
> | :----: | :----: | :----: | :----: | :----: |
> | LL2-7B-base | 4.5 | 10.9 | 6.1 | 3.7 |
> | LL3-8B-base | 5.7 | 10.8 | 4.4 | 3.8 |
>
> Due to the absence of instruction tuning, the base models struggle to follow instructions effectively. We will include this aspect in the ablation study.
>
> 2. As discussed in point 1, the base models often fail to adhere to the expected output patterns, making it difficult to identify and sample errors. To address this, we use Llama-3-8B-Instruct to generate the original reasoning chains for a small SFT dataset, S_{CS}, derived from the training data, which is then used to initialize the base models. It is important to note that the base models learn meta-skills (such as instruction-following abilities) from S_{CS} , rather than final reasoning capabilities, similar to the initial learning phase of Self-Evolution. After this step, we can perform Backtracking Correction without the need for additional data or annotations, since the pairwise training data is constructed during the self-correction process, without relying on annotations from closed-source models (e.g., GPT-3.5, GPT-4, etc.).
>
> 3. We will correct it in our paper.
>
> 4. During Backtracking Correction, we sample the errors and corrected answers dynamically. In other words, the sampled paths represent the actual mistakes the model makes at its current state. For each sample (q, D), although there may be multiple potential paths where the model could error, only the path that the model actually generates will be sampled. As a result, the addition of more documents contributes to a longer context length in the data, rather than causing an exponential increase in the time complexity of the training process. We will clarify this in Figure 2.

---

> > ### Comment · Reviewer_ofNF · 2024-11-27
> >
> > Thank you for your response.

---

### Official Review · Reviewer_gD7f · 2024-11-04

**Soundness:** 3
**Presentation:** 2
**Contribution:** 3
**Rating:** 5
**Confidence:** 3

**Summary:**

This paper introduces Backtracking Correction (BC), a novel approach for training large language models (LLMs) to detect noisy retrieved documents.

Backtracking Correction reformulates the task as a multi-step decision-making process, optimizing the model by iteratively correcting errors from the final reasoning steps back to the initial steps, thereby improving the model's ability to assess the relevance of retrieved documents and enhance the robustness of RAG against noisy inputs.

Experimental evaluations on multiple open-domain question-answering datasets demonstrate that Backtracking Correction outperforms multiple baselines without relying on additional data or extensive annotations.

**Strengths:**

+  The proposed Backtracking Correction method offers a novel approach to training LLMs for RAG tasks without relying on extensive external annotations, which could potentially reduce the cost and complexity of developing such systems.

**Weaknesses:**

+ The paper needs additional ablation studies and analysis for the proposed algorithm, especially for the efficiency claim and the scaling problem in term of chain length.
+ The experimental results are not consistently significant, with three out of four benchmarks (excluding SQuAD) showing either comparable performance or underperformance relative to baselines.

**Questions:**

+ Does BC implied a conditional depedency between the relevancy of the retrieved documents?
+ Could you provide the exact performance numbers for figure 3?
+ Could you include an efficiency analysis comparing the BC method to the baselines?
+ In section 4.2, the author claim that BC does not use extra data or annotations. Can you elaborate. Is BS's backbone trained from scratch? Does the training process of BC still require label of the reasoning chain as training data?
+ Can you elaborate on the learned ability of "summarizing the consistent or conflicting points" from BC?
+ Can you elaborate on the following claim that "the training data and reference models in Backtracking Correction are dynamic, while those in DPO are static"?

---

> ### Author Response · Authors · 2024-11-18
> **Response to Reviewer gD7f (part 1)**
>
> Thanks for your advice. We will respond to your questions point by point.
>
> 1. Yes. To be more precise, BC implies a dependency between the strictness or looseness of the standards used to judge the relevance of each retrieved document. Consider the process of assessing three documents: in decoder-only architectures, if the model evaluates the first two documents using a loose standard, it is likely to classify the third document as "relevant,"—and vice versa. In other words, the standard used to assess the relevance of the current document is influenced by the standards applied to the previous ones in the context. This can be viewed as a form of few-shot learning in an in-context setting.
>
> 2. Thanks for your advice. The performance are listed below:
>
> |  | Pop QA | Trivia QA | Web Questions | SQuAD |
> | :----: | :----: | :----: | :----: | :----: |
> | LST | 52.9 | 63.8 | 37.1 | 41.7 |
> | MLST | 53.5 | 62.1 | 38.6 | 41.5 |
> | CST | 55.4 | 64.7 | 43.4 | 42.3 |
> | BC | 58.4 | 68.1 | 47.6 | 44.5 |
>
> We will redraw the diagram and provide the exact performance numbers.
>
> 3. The inference stage of our method is similar to CST (e.g., Chain-of-Notes), where all retrieved documents for a given query are analyzed within the same context. In contrast, LST (e.g., Self-RAG) processes each retrieved document separately. This results in a significant increase in the number of items to be processed, which limits the ability to fully leverage frameworks like vllm for speed optimization, despite having a shorter average context length. Below are the time efficiency (in seconds) for inference across four datasets, based on an A800 80GB system:
>
> |  | Pop QA | Trivia QA | Web Questions | SQuAD |
> | :----: | :----: | :----: | :----: | :----: |
> | LST (Self-RAG) | 2634 | 17120 | 5114 | 25910 |
> | BC | 776 | 5024 | 1360 | 7392 |
> We will put it in our appendix.
>
> 4. Yes. We begin with the **base models** (Llama-2-7B-base and Llama-3-8B-base), as used in previous works (e.g., Self-RAG, REAR, Chain-of-Notes). However, these base models exhibit **poor instruction-following performance**, often failing to adhere to the expected output patterns. This makes it difficult to identify and sample errors. To address this, we use Llama-3-8B-Instruct to **generate the original annotations for a small SFT dataset**, S_{CS} , which is derived from the training data and used to initialize the base models. It is important to note that the base models learn meta-skills (instruction-following abilities) from S_{CS}, rather than final reasoning capabilities, similar to the **initial learning phase** in Self-Evolution. Once this step is completed, we can perform Backtracking Correction without the need for additional data or annotations. In contrast, previous works typically train on a large SFT dataset, S_{LS} or S_{CS} , annotated by closed-source models (e.g., GPT-3.5, GPT-4, etc.), meaning the base models learn final reasoning abilities from these strong annotators. As mentioned in the Introduction, Backtracking Correction eliminates the need for external annotations while still achieving competitive performance compared to methods relying on proprietary LLM annotations.

---

> ### Author Response · Authors · 2024-11-18
> **Response to Reviewer gD7f (part 2)**
>
> 5. The ability to summarize consistent and conflicting points forms the foundation of self-correction. Here we show an example:
>
> Query: What are forbidden on michigan's mackinac island?
>
> Document: [Mackinac Island] Bicycles, roller skates/roller blades, carriages, and saddle horses are available for rent. An road follows the island's perimeter, and numerous roads, trails and paths cover the interior. M-185, the United States' only state highway without motorized vehicles, makes a circular loop around the island, closely hugging the shoreline. The island is the location of Mackinac Island State Park, which covers approximately 80 percent of the island and includes Fort Mackinac as well as portions of the island's historic downtown and harbor. No camping is allowed on the island, but numerous hotels and bed.
>
> Original Judge: According to the document, there is no explicit mention of what is forbidden on Mackinac Island. However, the document does mention that \"No camping is allowed on the island\" which implies that camping is forbidden. This document is relevant to the question because it provides information about the rules and regulations on Mackinac Island, specifically mentioning what is not allowed, which is camping. So, the document is relevant to the question.
>
> Corrected Judge: Summary of the document: The document provides information about Mackinac Island, including its transportation options, road network, and attractions such as Mackinac Island State Park and Fort Mackinac. It also mentions that camping is not allowed on the island, but there are numerous hotels and bed and breakfast options available. The document is irrelevant to the question ‘What are forbidden on Michigan's Mackinac Island’ because it mentions what is allowed on the island, such as renting bicycles, roller skates/roller blades, carriages, and saddle horses, and what is not allowed (camping), but does not explicitly state what is forbidden. The document's focus is on the island's attractions, transportation, and accommodations, rather than its restrictions. So, the document is irrelevant to the question.
>
> As shown in the example, the model misjudged the document at first. However, provided with the prompt in the pink box on page 7 with the label ‘irrelevant’ (which has been already involved in the dataset), the model finds a conflicting point – the document does not explicitly state what is forbidden. In this way, we can utilize the model’s summarization ability to correct the wrong judgement.
>
> 6. BC conducts iterative training while DPO does not. Given a training dataset, BC:
> (1) divides the data into several splits,
> (2) collect the errors and correct them on the first split,
> (3) construct pairwise responses for the current split,
> (4) train the model on the constructed data,
> (5) collect the errors and correct them on the next split,
> (6) repeat (3)-(5) until finish training on all splits.
>
> The errors and corrected answers are generated by the iteratively trained model. In other words, they are sampled from the model’s output during training, reflecting the current distribution of the model (hence, we refer to them as dynamic). In contrast, DPO involves a one-time training process on the entire dataset, meaning that all pairwise responses are constructed before training and remain fixed throughout. Similarly, the reference model in DPO is the original model, and its parameters are not updated alongside the target model during training (so we say it is static). However, in BC, the reference model is specifically the model trained on the previous split.
> We will clarify this distinction further in our paper.

---

> ### Author Response · Authors · 2024-11-29
> **Response to Reviewer gD7f**
>
> Dear Reviewer gD7f,
>
> We would like to sincerely thank you for the time and effort you have dedicated to reviewing our manuscript and for providing invaluable feedback.
>
> As this phase is drawing to a close, we would like to confirm whether our responses have adequately addressed your concerns. We submitted detailed responses a few days ago, and we hope that they have resolved any issues you raised.
>
> If you need any further clarification or have additional concerns, please feel free to contact us. We are more than happy to continue our communication and assist in any way we can.
>
> Best regards,
>
> Authors

---

### Meta-Review · Area_Chair_zzES · 2024-12-21

**Metareview:**

(a) Scientific Claims and Findings:
The paper proposes BC, a method that iteratively trains LLMs to detect and correct errors in stepwise reasoning for RAG tasks. By reformulating RAG as a backward optimization process, BC reduces noise sensitivity and eliminates reliance on large annotated datasets. Experimental evaluations show that BC outperforms existing baselines on benchmark datasets.

(b) Strengths:

- Introduces an innovative reformulation of RAG training through backward optimization.
- Reduces dependency on external annotations, lowering development costs.
- Experimental results suggest consistent improvements in handling noisy inputs.

(c) Weaknesses:

- Insufficient analysis of scalability to larger datasets or more complex reasoning tasks (e.g., multi-hop reasoning).
- Some experimental results, especially for certain datasets, show only marginal improvements over baselines.
- Missing ablations for computational efficiency and comparisons with different model backbones.
- The mathematical formulation and notations require better clarity and additional examples in the appendix.

(d) Decision: accept
The paper addresses a relevant problem and introduces a novel method with practical implications. While some results are incremental, the reduction in annotation dependency and the innovative training strategy merit recognition. Weaknesses in scalability and experimental diversity slightly temper the impact.

**Additional Comments On Reviewer Discussion:**

Summary of Points Raised and Responses:

- Experimental Setup: Concerns about baseline fairness and computational efficiency were raised by multiple reviewers. The authors clarified that BC uses different initialization strategies compared to prior methods, with performance and efficiency metrics now included in the appendix.
- Scalability: Questions regarding BC's applicability to larger datasets and complex tasks were raised. The authors acknowledged the limitation, framing it as future work.
- Ambiguity and Edge Cases: Reviewers highlighted issues with ambiguous document relevance. Authors provided detailed case analyses and discussed the challenges of human annotation bias.
- Algorithm Details: Reviewers requested clarification on mathematical derivations and the naming of "Backtracking." The authors added explanations and examples to the appendix, partially addressing concerns.

Weighing Points in the Decision: The authors adequately addressed most concerns with detailed experiments and supplementary material. While some scalability and clarity issues remain, the responses improved the overall completeness of the submission.

---

### Decision · Program_Chairs · 2025-01-22

Accept (Poster)